# Effects of Physical Education Playfulness on Academic Grit and Attitude toward Physical Education in Middle School Students in The Republic of Korea

**DOI:** 10.3390/healthcare11050774

**Published:** 2023-03-06

**Authors:** Heonsu Gwon, Jongseob Shin

**Affiliations:** 1Industry-Academia Cooperation Center, Yong In University, Yongin 17092, Republic of Korea; 2Department of Exercise Rehabilitation & Welfare, Gachon University, Incheon 21936, Republic of Korea

**Keywords:** middle school, physical education, playfulness, academic grit, physical education attitudes

## Abstract

The purpose of this study was to explore the effect of playability in secondary physical education classes in Korea on academic grit and attitudes toward physical education. A total of 296 middle school students located in Seoul and Gyeonggi-do, Korea were surveyed via simple random sampling. Data were analyzed via descriptive statistical analysis, confirmatory factor analysis, reliability analysis, correlation analysis, and standard multiple regression analysis. Three primary results were obtained. First, playfulness was found to have a significant positive effect on academic grit. Specifically, mental spontaneity positively and significantly affected academic passion (β = 0.400), academic perseverance (β = 0.298), and consistency of academic interest (β = 0.297). Additionally, among the sub-variables of playfulness, humorous perspective was found to have a positive significant effect on maintaining consistency of academic interest (β = 0.255). The second primary finding was that playfulness had a significant positive effect on classroom attitudes to physical education. Specifically, physical animation and emotional fluidity were found to positively and significantly affect basic attitudes (β = 0.290 and 0.330, respectively) and social attitudes (β = 0.398 and 0.297, respectively). Third, academic grit was found to have a significant positive effect on PE classroom attitudes. Specifically, academic passion was found to have a positive and significant effect on basic attitudes (β = 0.427) and social attitude (β = 0.358). The results imply that attitude toward school life can be improved through physical activity in secondary physical education classes.

## 1. Introduction

On average, students who engage in intense or appropriate physical activities for more than three days a week are found to have a higher satisfaction with life than those who do not engage in physical activities at all [1]. In addition, it has been reported that students who actively engage in sports activities are less likely to engage in deviant behavior such as school violence, delinquency, and gaming addiction, and are less likely to develop depression [2,3,4].

However, in Korean society, physical education (PE) continues to attract less attention compared to other classes [5]. In Korean society, education is linked to college admissions. This education involves the effective application of cognitive skills related to subjects including Korean language, English, and mathematics. The community has neglected PE, in turn leading to social problems such as the poor physical strength of Korean teenagers, excessive competition for entrance exams, school violence, and suicide [5].

In particular, in 2011, a 13-year-old middle-school student was bullied by another student and eventually committed suicide in an apartment. This incident shocked the Korean public [6]. The Korean government then recognized the importance of PE, announced the “Seven Practical Tasks for Eradicating School Violence”, and instigated “Middle School Sports Club Activities” [7]. This policy has received positive reviews from students, stating that it contributes to interpersonal relationships, self-esteem, and physical development [8,9]. However, education in Korean society continues to ignore school playgrounds and focuses on cognitive skills rather than PE [10].

Above all, adolescence is a stage wherein individuals transition from childhood to adulthood. It directly affects physical, mental, and social development; if children grow up in a negative way, their societal lives will be characterized by negative thoughts and behaviors after they reach adulthood [11]. From this perspective, PE in adolescence can have a positive effect on physical, mental, and social growth; thus, adolescents should be encouraged to maintain continuous and diverse participation in PE [12].

Physical activity involves performing activities with pleasure [13]. Play is an important part of physical activity. Humans enjoy play from childhood onwards [14]. Since sports comprise elements of play, humans continue to participate in sports. From this perspective, elements of PE also include playability [15]. With continued participation in PE, individuals can develop patience and passion to achieve their goals [16]. During this process, individuals’ attitudes towards PE are also positively affected [17].

The concept of grit is important in physical activity. Grit is defined as exerting one’s potential through patience and passion for achievement of long-term goals [17], including continued long-term efforts despite the accompanying difficulties such as failure, frustration, and so on, that arise when achieving an objective [18]. High grit means that individuals are passionate about participating in PE and do not give up on difficult tasks, overcoming them with persistence and sincerity. In other words, if one feels pleasure toward PE through a positive experiences, academic grit is indicated, which positively affects PE-related attitudes, resulting in positive effects on life throughout adolescence.

Accordingly, studies on playability, academic grit, and classroom attitude have been conducted for psychological variables such as passion, positive thinking, and happiness [19,20,21,22,23,24,25,26], alongside studies related to sociality including academic results, school-life satisfaction, and interpersonal relationships [27,28,29,30,31,32]. However, there has been no study to date on how playability as an independent variable influences PE-related attitudes, with grit serving as a mediator.

Therefore, this study aimed to assess how playability in secondary PE classes affects middle-school students’ academic grit and PE-related attitudes. The research hypotheses are as follows: First, playability in secondary PE classes will affect academic grit. Second, playability in secondary PE classes will affect the attitudes of students. Third, academic grit will affect students’ attitudes toward PE.

### 1.1. Relationships between Playfulness, Academic Grit, and PE-Related Attitude

Playfulness positively affects adolescents’ academic achievement and life satisfaction through aspects such as positive emotions and pleasant experiences, motivation to laugh and act, promotion of creativity and human relationships, and relaxation of tension [19,20]. In particular, school sports are based on playability in order to develop pleasurable physical activities for participating individuals [15]. In other words, when playability expands, emotions such as passion develop, and positive activities can be achieved.

The concept of grit was introduced by Duckworth (2016), implying a strong will or motivation to pursue to a conclusion the goals one has set [18,23]. Previous studies showed that cognitive factors such as intelligence have direct effects on academic excellence [33]. However, in recent years, psychological factors including non-cognitive attitudes, values, and motives have been emphasized, and their effects on academic excellence have become apparent. Among them, grit forms an important factor. Grit is one of the representative non-cognitive abilities that stimulate an individual’s level of motivation, achievement, patience, self-efficacy, and clear sense of a goal [34,35]. Therefore, grit predicts success and is expressed as tenacity, persistence, determination, passion, persistence, sincerity, consistency of effort, and a sense of long-term purpose necessary for achieving an individual’s potential.

Furthermore, attitude includes a tendency to respond continuously and consistently to a specific object and is an important preceding variable that predicts behavior [36]. Within attitude, classroom attitude is a tendency to think positively or negatively about one’s education, through which students’ level of class participation can be predicted [37]. PE-related classroom attitude reflects the mindset or stature of a student in relation to PE classes. In other words, it involves individuals’ values or thoughts about PE classes. According to the theory of planned behavior, behavior and attitude are closely related [36]. 

Each of the variables described above has a close influence on the others. Previous studies have focused on playfulness, academic grit, and PE-related attitude. Specifically, studies on playfulness have investigated its relationship with psychological variables such as passion, positive thinking, and happiness [19,20,21], as well as how playfulness affects social behavior [27,28]. It was found that studies related to adolescents and playability have been conducted concerning sports, exercise, and study [38,39,40]. Since a human’s basic desire is to pursue pleasure, behavior that lacks playability cannot be continuously performed; thus, elements of pleasure are essential to enable continuous and positive results. Consequently, it was shown that adolescents with static playability could derive positive results in their studies.

In addition, research on adolescents and grit has been conducted assessing psychological variables, such as positive thinking, efficacy, resilience, and depression [22,41,42], social variables [29,30], and career variables [43,44]. Among these, studies on academic grit have attempted to explain the relationship between various variables relating to learning, such as classroom attitude, school life satisfaction, and class satisfaction [43,44]. In addition, studies related to sports have been conducted [45,46,47], showing that passionate athletes are more positive in terms of behavior, more active, and perform at their best in the process. Students with positive academic grit can produce excellent results in all subject-related studies, as well as positive results in PE classes. It has been stated that students with strong emotional variables, including grit, can excel in PE classes even if they do not possess outstanding physical skills.

Moreover, most of the research on adolescents and classroom attitude has been related to academic studies. Previous studies were conducted on students’ educational attitudes [31,32], academic achievement [48], persistence, passion, sincerity, and instructional attitudes [25,26] in relation to teacher behavior. Likewise, there have been many studies on academic attitudes in PE classes, alongside studies on the effect of psychological variables on PE classroom attitude and how the latter affects academic studies [49,50]. Students with a positive attitude toward PE have demonstrated positive results in school life, studies, and interpersonal relationships; they do not give up on tasks and they produce positive results with passion because they possess positive thoughts in general.

The abovementioned variables thus form the basis for supporting theoretical explanations in our study.

### 1.2. Study Hypotheses and Model

The hypotheses of this study were based on the effects of the playfulness of middle-school PE on academic grit and PE attitudes in the Republic of Korea. The hypothetical model is illustrated in Figure 1.

Figure 1 shows each hypothetical path of the conceptual model. This includes H1, which is the playfulness of middle-school PE, positively affecting academic grit. Next, H2 is academic grit, which positively affects attitudes toward middle-school PE. Lastly, H3 is the playfulness of middle-school PE, positively affecting PE attitudes. 

## 2. Materials and Methods

### 2.1. Study Participants

Middle-school students in Seoul and Gyeonggi Province, representing major population centres in the Republic of Korea, were selected as the study population. A sample of 350 students was selected using simple random sampling. Specifically, considering the characteristics of each region, Seoul was divided into four regions, and one school was selected from each of 28 cities in Gyeonggi-do. After excluding 54 questionnaires with incomplete or missing responses, the data from 296 participants were included in the analysis. According to the G-Power program, the appropriate number of samples was found to be greater than 118 (actual power: 0.80). The general characteristics of the study participants are shown in Table 1.

### 2.2. Instruments

Based on previous studies and theory, a structured questionnaire was employed to achieve the study’s objectives. The questionnaire comprised 53 items, including 2 for demographic characteristics, 20 for playfulness, 23 for academic grit, and 8 for PE attitude.

Playfulness was assessed using the questionnaire developed by Staempfli (2007), after modifying and adapting the items for this study [21]. The playfulness scale consisted of five factors: four items for physical animation, four for social engagement, four for mental spontaneity, four for emotional fluidity, and four for humorous perspective. Each item was rated on a five-point Likert scale.

Academic grit was assessed using items developed by Duckworth and Quinn (2009) modified and adapted for use in this study [23]. The scale consisted of three factors: ten items for academic passion, five for academic perseverance, and seven for academic consistency of interest. Each item was rated on a five-point Likert scale.

PE attitude was assessed using the scale developed by Kenyon (1968) modified and adapted for use in this study [50]. The scale consists of two factors: four items for basic attitude and four for social attitude. Each item was rated on a five-point Likert scale. Questionnaire questions for variables were added to Appendix Table A1.

### 2.3. Validity and Reliability of the Instruments

The content validity of the structured questionnaire was evaluated by two Sociology of Sports professors and one Ph.D. researcher. In addition, confirmatory factor analysis (CFA) was performed using maximum likelihood and Cronbach’s α to evaluate the validity and internal consistency of the instrument.

CFA for playfulness showed fit indices of χ^2^ = 300, df = 155, TLI = 0.907, CFI = 0.912, SRMR = 0.064, and RMSEA = 0.079. CFA for academic grit showed fit indices of χ^2^ = 373, df = 185, TLI = 0.933, CFI = 0.946, SRMR = 0.056, and RMSEA = 0.083. CFA for PE attitude showed fit indices of χ^2^ = 73.1, df = 19, TLI = 0.911, CFI = 0.940, SRMR = 0.039, and RMSEA = 0.079. Reliability, as tested with Cronbach’s α, ranged from 0.746 to 0.954 for the parameters, indicating good reliability, see Table 2, Table 3 and Table 4.

### 2.4. Investigative Procedure

To achieve the objectives of this study, we visited the selected schools and requested their cooperation. Data were collected from May to June 2022, until 350 samples had been extracted. The survey took 25 min to complete. We explained to the participants the purpose and method of completing the questionnaire and asked them to complete it via self-reporting. We conducted in-person interviews based on the questionnaire for participants who had difficulty responding. 

### 2.5. Data Processing

The collected data were processed. First, a frequency analysis was performed using SPSS 24.0. Then, the validity and reliability of the instruments were tested via CFA and Cronbach’s α, respectively, using Jamovi 1.6.23 [51,52,53]. Finally, correlation analysis and standard multiple regression analysis were performed using SPSS 24.0, to achieve the study’s objectives. Statistical significance was set at 0.05.

## 3. Results

### 3.1. Effects of Playfulness in Middle-School PE on Academic Grit

#### 3.1.1. Correlations among Study Variables

Table 5 shows the correlations between the factors with unidimensionality, in order to examine their satisfaction with discriminant validity. The correlation coefficients (r) among the associated variables ranged from 0.177 to.798, and there were partially significant correlations among the variables. However, the correlation coefficients did not exceed 0.80. Therefore, discriminant validity was deemed to be established according to the Kline (2011) criteria [54]. Moreover, the multicollinearity criterion was met (<0.80), confirming the absence of multicollinearity among the independent variables [55].

#### 3.1.2. Effects of Playfulness in Middle School PE on Academic Grit

Table 6 shows the results of the standard multiple regression performed to examine the effects of the playfulness of middle-school PE on the components of academic grit: academic passion, academic perseverance, and consistency of academic interest. 

The playfulness of middle-school PE explained 17.3% (*R*^2^_adj_ = 0.173) of the academic passion component of grit. Moreover, the mental spontaneity component of playfulness (β = 0.400) significantly affected academic passion (*p* = 0.000).

The playfulness of middle-school PE explained 11.7% (*R*^2^_adj_ = 0.117) of the academic perseverance component of academic grit. Moreover, the mental spontaneity component of playfulness (β = 0.298) significantly affected academic performance (*p* = 0.003).

The playfulness of middle-school PE explained 13.0% (*R*^2^_adj_ = 0.130) of the consistency of academic interest component of academic grit. Moreover, the mental spontaneity (β = 0.297) and humorous perspective (β = 0.255) components of playfulness significantly affected consistency of academic interest (*p* = 0.003 and 0.050, respectively).

### 3.2. Effects of Playfulness in Middle-School PE on PE Attitude

Table 7 shows the results of the standard multiple regression examining the effects of playfulness in middle-school PE on the basic and social components of PE attitude. 

The playfulness of middle-school PE explained 29.8% (*R*^2^_adj_ = 0.298) of the basic component of PE attitude. Moreover, the physical animation (β = 0.290) and emotional fluidity (β = 0.330) components of playfulness had significant effects on basic attitude (*p* = 0.003 and 0.000, respectively). 

The playfulness of middle-school PE explained 29.9% (*R*^2^_adj_ = 0.299) of the social component of PE attitude. Moreover, the physical animation (β = 0.398) and emotional fluidity (β = 0.297) components of playfulness had significant effects on social attitude (*p* = 0.000 and 0.001, respectively).

### 3.3. Effects of Academic Grit on PE Attitude

Table 8 shows the results of the standard multiple regression analysis examining the effects of academic grit on the basic and social components of PE attitudes in middle school.

Academic grit explained 18.3% (*R*^2^_adj_ = 0.183) of the basic component of PE attitudes. Moreover, the academic passion component (β = 0.427) of academic grit significantly affected basic attitudes (*p* = 0.019). 

Academic grit explained 13.7% (*R*^2^_adj_ = 0.137) of the social component of PE attitudes. Moreover, the academic passion component (β = 0.358) of academic grit significantly affected social attitudes (*p* = 0.050).

## 4. Discussion

This study aimed to investigate the effects of playfulness in middle-school PE on academic grit and PE attitudes. The study sample was drawn from middle-school students in Seoul and Gyeonggi Province, South Korea, and data from 296 participants were analyzed.

First, we examined the effects of playfulness in middle-school PE on academic grit. Playfulness positively affected academic passion, perseverance, and consistency of interest, associated with academic grit. Other studies have also reported similar results regarding the playfulness of PE classes and academic grit [47,55,56,57,58].

Barnett (1990) stated that playfulness is a personality trait of humans and represents the degree to which a person freely expresses pleasure or joy [59]. In other words, playfulness is an internal pleasure and happiness that one expresses. Playfulness is essential for humans who cannot persistently engage in a behavior where play is lacking [60]. Therefore, playfulness is essential to humans. Playfulness may be displayed in all patterns of behavior, in addition to leisure activities. All activities, including studying, music, art, and exercise, encompass playfulness, and only people who enjoy these activities perform well in them [19]. Some people achieve good academic performance, while others excel at sports; the difference is attributable to the different areas that people enjoy as play. The discrepancy of outcome is due more to how one enjoys that particular activity than to one’s in-born competency [20]. 

Playfulness and academic grit are believed to have similar properties [61]. Several studies have argued that these two factors are strongly associated [38,39,40], with people who strongly exhibit playfulness also showing high academic grit [39]. Similarly, students who demonstrated a high degree of playfulness in middle-school PE showed higher academic grit. In other words, students who perceive playful features in PE classes more frequently and strongly display an unwavering passion for academic pursuits. Furthermore, these students can be expected to acheive positive outcomes in academic study and also PE. 

Based on our findings, instructors should research ways to engage in PE classes so that male and female students can enjoy PE. In addition, teachers must strive to help students feel they can enjoy their studies. If students can enjoy class, they will be more engaged and achieve better academic outcomes. Moreover, these students will be able to produce consistent results based on their high confidence levels. This will positively affect academic grit, leading to positive outcomes in their overall lives. 

Secondly, we analyzed the effects of academic grit on PE attitudes among middle-school students. The results showed that academic grit positively affected the basic and social components of PE attitudes. More specifically, academic grit has a similar property to passion, so people with strong academic grit are passionate during PE classes, as indicated by many studies [25,48,62]. In addition, people with academic grit show positive attitudes in other academic areas besides PE, and engage in positive school lives, interpersonal relationships, and social behaviors [45,46].

Grit is the display of perseverance and passion to overcome adversity and failure to achieve a long-term goal [63]. Similarly, academic grit refers to the passion, perseverance, and effort to achieve academic goals. In other words, it focuses on non-cognitive attitudes and behaviors, as opposed to cognitive features, required to attain good grades in school [63]. Even if students have poor cognitive abilities, students with strong academic grit attain good grades more often than students with good cognitive abilities that do not strive to achieve their goals of good academic performance. Students with strong academic grit do not fear failure and continue to advance toward their ultimate goals.

This attitude is also displayed in PE, where students with better physical abilities ultimately attain better outcomes and enjoy class more than their less capable counterparts [64,65]. Those students with poorer physical abilities fear PE classes and may want to remain passive. Such an attitude leads to their experiencing physical activities unpleasantly, which ultimately discourages them from exercise for the rest of their lives. However, students with strong academic grit do not give up during PE and continue to work hard until they accomplish the given task and eventually succeed [25,32,44,45,46,48]. An accumulation of positive experiences like this instills good memories of exercise and motivates individuals to engage in sports throughout their lives. Moreover, they are better able to seize other opportunities in life, perpetuating a virtuous cycle. 

In other words, fostering grit in more students may help nurture those who are not merely focused on competing and winning against others—a mindset rampant in public schools in Korea—and can instead positively influence society [66,67]. Thus, diverse types of education, social systems, and educational programs that foster emotional abilities such as passion, hard work, perseverance, and grit, should be developed instead of focusing only on improving cognitive abilities. Continued advances based on positive processes and results will transform our society and help to resolve, at least partially, societal problems that are prevalent worldwide. 

Third, we investigated the effects of playfulness in middle-school PE on PE attitudes. The results showed that playfulness positively affected the basic and social components of PE attitudes. Specifically, students who perceived more playfulness and enjoyment in PE displayed more active attitudes during PE, as observed in many studies [31,32,68].

Playfulness is an internal tendency that enables high-quality playful interactions in various environments and contexts [11]. Students who lack playfulness fail to perceive the playful features of the activities and tasks presented by teachers or other students and consider them as assignments. This increases their vulnerability to psychological pressures and stress. On the other hand, students with a high level of playfulness easily detect playful components in various aspects of their lives and readily discover fun things in life [38]. Hence, they reflect on their desires or thoughts through play and gain a happy experience. Previous studies on playfulness and academic activity support our findings that students’ playfulness positively influences their classroom attitudes [39]. That is, students with a high level of playfulness discover playful features in various situations at school, can effectively cope with academic stress or psychological pressure, and are more likely to enjoy happy and fun lives at school.

The playfulness of school PE and PE attitudes among adolescents were found to be significantly interrelated. Playfulness positively influences PE attitude and academic grit, and this result sheds light on the desired direction on which school PE should focus. Being playful is associated with high energy and being physically and socially active. Playful individuals can actively respond to a threatening or critical situation without perceiving it negatively [60,68,69,70]. Our findings showed that adolescents who enjoy PE could enhance their physical fitness and be motivated to participate in all activities, maintaining a positive attitude. As such, perceived pleasure manifests as voluntary involvement in class. Here, activity level is a positive factor that improves mental health through physical development [71]. Students’ attitudes toward PE are crucial because they focus on the process (participating or not in physical activity during PE) and the consequent outcomes, as opposed to being results-oriented and making students overly competitive while focusing primarily on their grades.

Therefore, instructors must advance their PE classes by experimenting with various approaches instead of simply adopting and following the prevailing trend. Moreover, they should not divide students based on “good“ or “bad“ performances in PE. Instead, they should implement physical activity education that stresses playfulness, so that all students can enjoy PE and feel motivated to seek challenges. Adolescents who learn physical activity while enjoying it will acquire passion, a challenge-seeking mindset, and a persistent attitude that encourages them never to give up. If such practices are established as habits in daily life, they will be able to paint a more promising future for themselves.

## 5. Conclusions

This study aimed to investigate the effects of playfulness in middle-school PE on academic grit and PE attitude. We drew the following conclusions based on our results: the playfulness of middle-school PE significantly affected academic grit, academic grit significantly affected PE attitudes, and the playfulness of middle-school PE significantly affected PE attitudes. Finally, these results show that engaging in physical activity through middle-school PE can enhance students’ attitudes toward their education. 

There were a few limitations to this study. First, the research was conducted focusing on the relationship between variables, analyzed through regression analysis; therefore, there was limited analysis of differences between genders and school grades, among demographic characteristics. If the study had focused on the relationship between variables in terms of differences between genders or grade levels, the results would be interpreted differently. Second, this study is limited in terms of its investigation of the effect of individual psychology on variables such as playfulness, academic grit, and attitude toward PE. Future studies should focus on how PE in school affects our society, looking at the relationships between macroscopic variables. Third, in the regression analysis, there were no controls for gender, race, family characteristics, teacher characteristics, or school characteristics; however, it is possible that these variables may affect attitudes toward PE and academic grit in students.

Future research should focus on examining PE attitudes based on a fundamental scale for games and sports, as this will shed light on the practical effects of physical activity. Moreover, although we examined attitudes toward PE, students’ attitudes in other academic areas should also be examined. Thus, student participation in school sports clubs should also be investigated to explore measures that promote values and positive attitudes toward sports.

## Figures and Tables

**Figure 1 healthcare-11-00774-f001:**
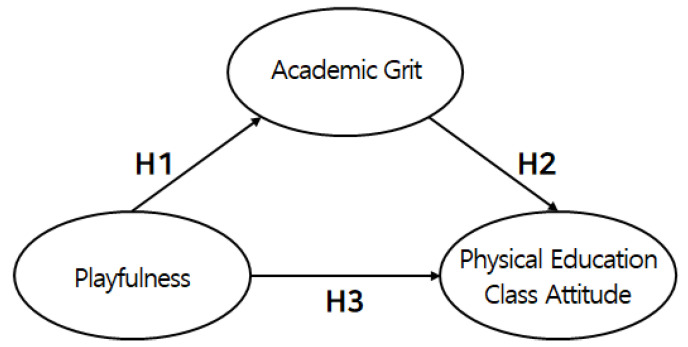
Study model.

**Table 1 healthcare-11-00774-t001:** Participants’ general characteristics.

Variable		Frequency (n)	Percentage (%)
Sex	Male	170	57.4
	Female	126	42.6
Grade Level	6th grade	72	24.3
	7th grade	192	64.9
	8th grade	32	10.8

**Table 2 healthcare-11-00774-t002:** CFA and reliability of school happiness.

Variable	Latent Variable	Measurement Variable	B	β	S.E.	t	α
Playfulness	Physical animation	a01	1	0.585			0.799
		a02	1.041	0.651	0.174	5.97 ***	
		a03	1.380	0.814	0.195	7.08 ***	
		a04	1.289	0.810	0.182	7.08 ***	
	Social engagement	b01	1	0.542			0.766
		b02	1.227	0.691	0.198	6.20 ***	
		b03	1.484	0.764	0.231	6.44 ***	
		b04	1.432	0.694	0.231	6.20 ***	
	Mental spontaneity	c01	1	0.662			0.746
		c02	0.560	0.419	0.142	3.95 ***	
		c03	0.785	0.618	0.152	5.18 ***	
		c04	0.705	0.503	0.142	4.98 ***	
	Emotional fluidity	d01	1	0.677			0.805
		d02	1.054	0.781	0.140	7.52 ***	
		d03	0.863	0.677	0.124	6.96 ***	
		d04	1.027	0.718	0.140	7.35 ***	
	Humorous perspective	e01	1	0.701			0.853
		e02	1.215	0.775	0.138	8.84 ***	
		e03	1.321	0.848	0.137	9.62 ***	
		e04	1.102	0.764	0.133	8.30 ***	

*** *p* < 0.001.

**Table 3 healthcare-11-00774-t003:** CFA and reliability of academic grit.

Variable	Latent Variable	Measurement Variable	B	β	S.E.	t	α
Academic grit	Academic passion	a01	1	0.922			0.954
		a02	1.049	0.909	0.054	19.14 ***	
		a03	1.058	0.947	0.048	21.89 ***	
		a04	0.967	0.825	0.065	14.84 ***	
		a05	0.812	0.751	0.066	12.18 ***	
		a06	0.893	0.812	0.063	14.18 ***	
		a07	0.960	0.845	0.061	15.55 ***	
		a08	0.841	0.734	0.071	11.69 ***	
		a09	0.774	0.666	0.078	9.88 ***	
		a10	0.690	0.663	0.070	9.84 ***	
	Academic perseverance	b01	1	0.814			0.933
		b02	1.127	0.825	0.095	11.83 ***	
		b03	1.145	0.885	0.088	12.96 ***	
		b04	1.073	0.850	0.088	12.12 ***	
		b05	1.134	0.841	0.094	12.07 ***	
	Consistency of academic interest	c01	1	0.723		7.25 ***	0.924
		c02	0.758	0.696	0.104	9.28 ***	
		c03	1.015	0.759	0.109	11.32 ***	
		c04	1.244	0.923	0.109	11.32 ***	
		c05	1.271	0.922	0.112	8.60 ***	
		c06	0.903	0.711	0.105	11.08 ***	
		c07	1.200	0.904	0.108	7.25 ***	

*** *p* < 0.001.

**Table 4 healthcare-11-00774-t004:** CFA and reliability of attitude toward PE.

Variable	Latent Variable	Measurement Variable	B	β	S.E.	t	α
PE attitude	Basic attitude	a01	1	0.848			0.923
		a02	0.985	0.860	0.073	13.40 ***	
		a03	1.077	0.894	0.075	14.22 ***	
		a04	1.002	0.859	0.074	13.42 ***	
	Social attitude	b01	1	0.751			0.840
		b02	0.981	0.769	0.102	9.62 ***	
		b03	1.159	0.872	0.109	10.60 ***	
		b04	0.958	0.632	0.126	7.60 ***	

*** *p* < 0.001.

**Table 5 healthcare-11-00774-t005:** Correlations among playfulness, academic grit, and attitude toward PE.

		1	2	3	4	5	6	7	8	9	10	Skewness	Kurtosis	M (SD)
1	Physical animation	1										−0.103	−0.463	3.56 (0.849)
2	Social engagement	0.695	1									−0.138	−0.486	3.77 (0.801)
3	Mental spontaneity	0.378	0.519	1								0.269	−0.431	3.54 (0.736)
4	Emotional fluidity	0.387	0.550	0.459	1							−0.102	−0.974	3.81 (0.807)
5	Humorous perspective	0.565	0.762	0.575	0.568	1						0.094	−0.406	3.35 (0.954)
6	Academic passion	0.182	0.247	0.443	0.259	0.296	1					0.002	0.058	3.15 (0.946)
7	Academic perseverance	0.184	0.249	0.367	0.183	0.299	0.773	1				0.110	0.158	3.08 (0.974)
8	Consistency of academic interest	0.138	0.227	0.357	0.125	0.312	0.729	0.706	1			0.009	−0.065	2.93 (0.926)
9	Basic attitude	0.447	0.427	0.369	0.477	0.371	0.312	0.257	0.220	1		−0.375	−0.406	3.82 (0.878)
10	Social attitude	0.491	0.407	0.328	0.445	0.375	0.210	0.132	0.177	0.798	1	−0.389	−0.168	3.90 (0.823)

**Table 6 healthcare-11-00774-t006:** Standard multiple regression for the effects of playfulness on academic grit.

Variable	Academic Passion		Academic Perseverance		Consistency of Academic Interest		VIF
	β	t	β	t	β	t	
Physical animation	0.000	0.005	0.002	0.016	−0.064	−0.601	1.944
Social engagement	−0.043	−0.314	0.002	0.014	−0.006	−0.046	3.320
Mental spontaneity	0.400	4.254 ***	0.298	3.069 **	0.297	3.074 **	1.573
Emotional fluidity	0.064	0.674	−0.039	−0.402	−0.127	−1.311	1.597
Humorous perspective	0.062	0.492	0.147	1.134	0.255	1.979 *	2.811
*adjusted R* ^2^ *F*	*adjusted R*^2^ = 0.173 *F* = 7.167 ***		*adjusted R*^2^ = 0.117 *F = 4.897 ****		*adjusted R*^2^ = 0.130 *F* = 5.380 ***		

* *p* < 0.05, ** *p* < 0.01, *** *p* < 0.001.

**Table 7 healthcare-11-00774-t007:** Standard multiple regression for the effects of playfulness on attitude toward PE.

Variable B	Basic Attitude		Social Attitude		VIF
	β	t	β	t	
Physical animation	0.290	3.007 **	0.398	4.137 ***	1.944
Social engagement	0.041	3.28	−0.061	−0.488	3.320
Mental spontaneity	0.140	1.614	0.085	0.978	1.573
Emotional fluidity	0.330	3.786 ***	0.297	3.407 ***	1.597
Humorous perspective	−0.092	−0.796	−0.020	−0.177	2.811
*adjusted R* ^2^ *F*	*adjusted R*^2^ = 0.298 *F* = 13.503 ***		*adjusted R*^2^ = 0.299 *F = 13.542 ****		

** *p* < 0.01, *** *p* < 0.001.

**Table 8 healthcare-11-00774-t008:** Standard multiple regression for the effects of academic grit on attitude toward PE.

Variable	Basic Attitude		Social Attitude		VIF
	β	t	β	t	
Academic passion	0.427	2.377 **	0.358	1.942 *	5.173
Academic perseverance	−0.022	−0.130	−0.239	−1.375	4.624
Academic consistency of interest	−0.116	−0.781	0.073	0.483	3.523
*adjusted R* ^2^ *F*	*adjusted R*^2^ = 0.183 *F* = 5.443 ***		*adjusted R*^2^ = 0.137 *F = 2.865 **		

* *p* < 0.05, ** *p* < 0.01, *** *p* < 0.001.

## Data Availability

The data presented in this study are available on request from the corresponding authors.

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
