# Peer review of "Effects of Physical Education Playfulness on Academic Grit and Attitude toward Physical Education in Middle School Students in The Republic of Korea"

_healthcare, 2023, doi:10.3390/healthcare11050774_

Round 1
Reviewer 1 Report
This study entitled “Effects of Physical Education Playfulness on Academic Grit and Attitude Toward Physical Education in Middle School Students in The Republic of Korea” aimed to investigate the effects of playfulness in middle school PE on academic grit and PE attitudes.
The research has scientific relevance. The findings were novel. It serves a useful and meaningful contribution to the field, especially that the content of the study is unique to Middle School Students.
However, the authors have to respond some comments regarding the manuscript.
Recommendations for some revisions are outlined below.
- A sample of 350 students 167 was selected using simple random sampling. à Could you describe more the procedure for recruiting students ?
- Try to explain if you asked for more than 296 students? and how did you arrive at this number of participants?
- Could you describe more the study participants Section.
- The time required to fill out the questionnaire was not mentioned. Please mention it.
- You mentioned in Table 1 that your participants are divided into 170 male and 126 female. Have you found a sex effect on your variables ? If so, why didn’t you mention them in your manuscript ? A comparison between male and female would be interesting and informative to your study.
- Table 5 needs to be reviewed : the first column numbers need to be aligned.
- Please harmonize spacing for Table 7 as for other tables.
- In your opinion, this study had limits or not? If so, mention them in a Limits section.
Author Response
Response to Reviewer 1
We appreciate for your insightful and useful comments. We’ve revised to reflect your comment.
Point 1: A sample of 350 students 167 was selected using simple random sampling. à Could you describe more the procedure for recruiting students ?
Response 1: As requested, the contents have been modified and supplemented. I added it to the study participants section of 2.1. The contents below are modified. Specifically, considering the characteristics of each region, Seoul was largely divided into four regions, and one school was selected out of 28 cities in Gyeonggi-do.
Thank you for your comments and suggestions.
Point 2: Try to explain if you asked for more than 296 students? and how did you arrive at this number of participants?
Response 2: As requested, the contents have been modified and supplemented. I added it to the study participants section of 2.1. The contents below are modified. As a result of the G-Power program, the appropriate number of samples for sampling was found to be more than 118 (actual power: .80).
The general characteristics of the study participants are shown in Table 1.
Point 3: Could you describe more the study participants Section.
Response 3: Thank you for your valuable comments and suggestions. When administering the survey, the general characteristics of the study participants were limited to gender and grade. The focus of the study was on regurgitation analysis rather than t-test and F-testing according to gender or grade. Therefore, we did not analyze any other characteristics except for these two items. We hope for your understanding.
Point 4: The time required to fill out the questionnaire was not mentioned. Please mention it.
Response 4: As requested, the contents have been modified and supplemented. I added it to the investigative procedure section of 2.4. The contents below are modified. Data were collected from May to June 2022 until 350 samples were extracted. It took 25 minutes to complete the survey.
Point 5: You mentioned in Table 1 that your participants are divided into 170 male and 126 female. Have you found a sex effect on your variables ? If so, why didn’t you mention them in your manuscript ? A comparison between male and female would be interesting and informative to your study.
Response 5: Thank you for your valuable suggestions. The reason for dividing males and females in Table 1 was not to analyze the effect of gender, but to derive a result that is not biased toward a specific gender. The comparative study between males and females that you mentioned will be conducted as a follow-up study. This is because males and females have different attitudes toward physical education in the Korean society.
Point 6: Table 5 needs to be reviewed : the first column numbers need to be aligned.
Response 6: As requested, the contents have been modified and supplemented. The contents below are modified. I revised it to reflect the reviewer's opinion.
Point 7: Please harmonize spacing for Table 7 as for other tables.
Response 7: As requested, the contents have been modified and supplemented. The contents below are modified. I revised it to reflect the reviewer's opinion. Thank you for your comments and suggestions.
Point 8: In your opinion, this study had limits or not? If so, mention them in a Limits section.
Response 8: As requested, the contents have been modified and supplemented. I added to the 5. conclusions section. The contents below are modified.
This study focused on the relationship between variables through regression analysis. therefore, there is a limit to the analysis of differences between men and women and grades among demographic characteristics. if you look at the relationship between variables through the difference between men and women and the difference between grades, a different interpretation from this study will appear. in addition, this study has limitations in the importance of individual psychology such as playfulness, academic grit, and attitude toward PE. other studies have studied macroscopically. we studied how PE in school affects our society. looking at the relationship between macroscopic variables, meaningful results will appear for PE.
Thank you for your comments and suggestions.
For more details please see the revised version manuscript.

Reviewer 2 Report
row 168- "After excluding 54 questionnaires with careless or missing responses" How were the excluded 54 questionnaires distributed across the study arms? i.e. could you see missing responses in all items (playfulness, academic grit, and PE attitude) or just in one field?
Author Response
Aappreciate for your insightful and useful comments.
We’ve revised to reflect your comment.
Please refer to the attached file.
Thank you for your comments and suggestions.
--
Response to Reviewer 2
We aappreciate for your insightful and useful comments. We’ve revised to reflect your comment.
Point 1: 1. row 168- "After excluding 54 questionnaires with careless or missing responses" How were the excluded 54 questionnaires distributed across the study arms? i.e. could you see missing responses in all items (playfulness, academic grit, and PE attitude) or just in one field?
Response 1: As requested, the contents have been modified and supplemented. The contents below are modified. Thank you for your valuable comments. The insincere responses in a number of surveys were abandoned.
The survey took more than 25 minutes. Some of the middle school students seemed to be bored. Therefore, after the first item on playfulness, the remaining survey was not answered in some cases. Thank you for your comments and suggestions.
For more details please see the revised version manuscript.

Reviewer 3 Report
I appreciate the opportunity to review this work. The purpose of this paper is to investigate the effects of middle school physical education (PE) playfulness on academic grit and PE attitudes among middle school students in the Republic of Korea. I have a few comments and questions and in particular, I think this study should be better motivated and the contributions should be more clearly stated; the data and methods should also be better justified. My detailed comments are as follows:
1. In the abstract, you do not need to provide software information. Instead, detailed results of major findings should be elaborated.
2. Line 32, “…lays a foundation to foster students into adults who remain physically active” this sentence is odd.
3. Line 53, Academics are directly linked to adolescents’ futures in most cases. why and how? The authors need have citations here.
4. The introduction is not well structured and looks too long. For me, it is more like a literature review. I would strongly suggest the authors include a better motivation for this study, some Korean school background for international audiences who are not familiar with Korean education, and briefly discuss what has been and has not been found in the existing literature. The authors also need to point out what’s the contributions of this study, both theoretical and practical. Then, many of the contexts in the current introduction could be included in Section 2. Literature Review to build their rationales for the hypotheses model.
5. Section 2.2., the authors need to explain how items were measured like 1= strongly disagree and 5 = strongly agree. The authors also need to list all the survey items in an appendix to tell readers what the survey questions are, and whether they are reasonable, as well as for transparency.
6. For survey items, generally the answers are not normally distributed, they are skewed, so the authors need to provide skewness and kurtosis, and use WLSMV, rather than maximum likelihood.
7. In section 2.3., based on the reported fit indices, the CFA model does not fit the data well so the authors need to adjust their CFA model. Based on Cronbach’s alpha, authors could not conclude good reliability since the smallest value of Cronbach’s alpha is 0.713. However, in the tables, I did not see the number 0.713, the authors do not report their results correctly, and need further checking.
8. It is better to combine to Table 2, 3, and 4 together, and change the title accordingly. Then number the latent variable as the authors did in Table 5.
9. Table 5 missed one column. There is no need to report significance.
10. The authors conducted CFA, and correlation analysis, why the authors do not construct an SEM. Instead, they use regression models. I am not sure whether they conduct multiple regressions only by two variables for each regression, or examine jointly. The authors need to clarify this.
11. For the discussion, limitations should also be pointed out.
Author Response
Dear Reviewer,
We appreciate for your insightful and useful comments. We’ve revised to reflect your comment.
Point 1:
In the abstract, you do not need to provide software information. Instead, detailed results of major findings should be elaborated.
Response 1:
As requested, the contents have been modified and supplemented. I modified it to abstract.
The contents below are modified.
Abstract: This study aimed to explore the effect of playability in physical education classes in Korea on academic grit and attitude toward physical education among middle school students. Random sampling was conducted with 296 middle school students in Seoul and Gyeonggi-do. Descriptive statistical analysis, confirmatory factor analysis, reliability analysis, correlation analysis, and standard multiple regression were performed with the data. Results showed that playfulness had a significant effect on academic grit in secondary physical education classes. Specifically, mental spontaneity among the sub-variables of playfulness had a significant effect on academic passion, academic perseverance, and academic consistency of interest. Additionally, among the sub-variables of playfulness, humorous perspective had a significant effect on maintaining academic consistency of interest. Second, there was a significant effect of playfulness on attitude toward physical education among students. Specifically, physical animation and emotional fluidity, among the subvariables of playfulness, had a significant effect on basic and social attitude. Third, academic grit had a significant influence on attitude toward physical education among middle school students. Specifically, academic grit had a significant effect on basic and social attitude. Thus, our findings show that attitude toward school life can be increased through physical activities in among middle school students.
Point 2:
Line 32, “…lays a foundation to foster students into adults who remain physically active” this sentence is odd.
Response 2:
As requested, the contents have been modified and supplemented.
Point 3:
Line 53, Academics are directly linked to adolescents’ futures in most cases. why and how? The authors need have citations here.
Response 3:
As requested, the contents have been modified and supplemented.
The introduction has been modified.
Thank you for your comments and suggestions.
Point 4:
The introduction is not well structured and looks too long. For me, it is more like a literature review. I would strongly suggest the authors include a better motivation for this study, some Korean school background for international audiences who are not familiar with Korean education, and briefly discuss what has been and has not been found in the existing literature. The authors also need to point out what’s the contributions of this study, both theoretical and practical. Then, many of the contexts in the current introduction could be included in Section 2. Literature Review to build their rationales for the hypotheses model.
Response 4:
As requested, the contents have been modified and supplemented.
I revised the introduction. and we modified the relationship between variables. The contents below are modified.
Point 5:
Section 2.2., the authors need to explain how items were measured like 1= strongly disagree and 5 = strongly agree. The authors also need to list all the survey items in an appendix to tell readers what the survey questions are, and whether they are reasonable, as well as for transparency.
Response 5:
As requested, the contents have been modified and supplemented. The contents below are modified. And I added the questionnaire to the appendix.
Point 6:
For survey items, generally the answers are not normally distributed, they are skewed, so the authors need to provide skewness and kurtosis, and use WLSMV, rather than maximum likelihood.
Response 6:
As requested, the contents have been modified and supplemented.
Point 7:
In section 2.3., based on the reported fit indices, the CFA model does not fit the data well so the authors need to adjust their CFA model. Based on Cronbach’s alpha, authors could not conclude good reliability since the smallest value of Cronbach’s alpha is 0.713. However, in the tables, I did not see the number 0.713, the authors do not report their results correctly, and need further checking.
Response 7:
As requested, the contents have been modified and supplemented. The contents below are modified.
It's 0.746 but it's a typo of 0.713. I modified it.
Point 8:
It is better to combine to Table 2, 3, and 4 together, and change the title accordingly. Then number the latent variable as the authors did in Table 5.
Response 8:
Thank you for your valuable suggestion. We agree that it is better to combine Tables 2, 3, and 4 together. However, since our idea was to show the CFA for each variable, segregating the tables would be more appropriate for the readers’ understanding. We have therefore left the tables accordingly.
Point 9:
Table 5 missed one column. There is no need to report significance.
Response 9:
Thank you for your comment. Kindly provide more information on the missing column that you are referring to since we are unable to pinpoint what you mean. As for the significance levels, we have deleted the asterisks and corresponding significance levels. Please check if this reflects your comment.
Point 10:
The authors conducted CFA, and correlation analysis, why the authors do not construct an SEM. Instead, they use regression models. I am not sure whether they conduct multiple regressions only by two variables for each regression, or examine jointly. The authors need to clarify this.
Response 10:
Thank you for your valuable comment This study was not analyzed through structural equation modeling. It is more meaningful to observe the influence between sub-variables than the fit of the model. Thus, we found it suitable to use multiple regression analysis to assess any meaningful influence of independent and dependent variables. Consequently, as shown in Figure 1, three research problems (H1 to H3) were presented.
Point 11:
For the discussion, limitations should also be pointed out.
Response 11:
As requested, the contents have been modified and supplemented. I added to the 5. conclusions section.
The contents below are modified.
This study focused on the relationship between variables through regression analysis. therefore, there is a limit to the analysis of differences between men and women and grades among demographic characteristics. if you look at the relationship between variables through the difference between men and women and the difference between grades, a different interpretation from this study will appear. in addition, this study has limitations in the importance of individual psychology such as playfulness, academic grit, and attitude toward PE. other studies have studied macroscopically. we studied how PE in school affects our society. looking at the relationship between macroscopic variables, meaningful results will appear for PE.
Thank you for your comments and suggestions.
For more details please see the revised version manuscript.

Round 2
Reviewer 3 Report
I appreciate the authors’ efforts in revising their manuscript. Substantial improvements were made. However, I still have some concerns, as follows.
1. In the abstract, the authors should point out the exact results. For example, the authors stated: “Results showed that playfulness had a significant effect on academic grit in secondary physical education classes.” So is it significantly positive or negative? And to what degree? The authors should illustrate the results clearly for readers in the abstract.
2. Formatting issues. The first column in Table 5 should be adjusted, better to display the content in a single line. For the questionnaire, do not display the content vertically. These are hard to read.
3. Line 369, avoid using “you” in a research article.
4. In the limitation, at least, the authors should mention that in their regression models, they do not include any controls such as gender, race/ethnicity, family characteristics, teacher characteristics, school characteristics, which are likely to also affect students’ PE attitude and academic grit.
5. English language needs to be further checked.
Author Response
As requested, the contents have been modified and supplemented.
Thank you for your comments and suggestions.
Response to Reviewer 3 Comments
We appreciate for your insightful and useful comments. We’ve revised to reflect your comment.
Point 1: In the abstract, the authors should point out the exact results. For example, the authors stated: “Results showed that playfulness had a significant effect on academic grit in secondary physical education classes.” So is it significantly positive or negative? And to what degree? The authors should illustrate the results clearly for readers in the abstract.
Response 1: the contents have been modified and supplemented. The contents below are modified.
Point 2: In the abstract, the authors should point out the exact results. For example, the authors stated: “Results showed that playfulness had a significant effect on academic grit in secondary physical education classes.” So is it significantly positive or negative? And to what degree? The authors should illustrate the results clearly for readers in the abstract.
Response 2: the contents have been modified and supplemented. The contents below are modified.
Point 3: Line 369, avoid using “you” in a research article.
Response 3: the contents have been modified and supplemented. The contents below are modified.
Point 4: In the limitation, at least, the authors should mention that in their regression models, they do not include any controls such as gender, race/ethnicity, family characteristics, teacher characteristics, school characteristics, which are likely to also affect students’ PE attitude and academic grit.
Response 4: the contents have been modified and supplemented.
Point 5: English language needs to be further checked
Response 5: the contents have been modified and supplemented. I checked English by requesting a correctional expert.
For more details please see the revised version manuscript.
